# Identification of Potential microRNA Panels for Male Non-Small Cell Lung Cancer Identification Using Microarray Datasets and Bioinformatics Methods

**DOI:** 10.3390/jpm12122056

**Published:** 2022-12-13

**Authors:** Antonia Haranguș, Raduly Lajos, Livia Budisan, Oana Zanoaga, Cristina Ciocan, Cecilia Bica, Radu Pirlog, Ioan Simon, Marioara Simon, Cornelia Braicu, Ioana Berindan-Neagoe

**Affiliations:** 1Research Center for Functional Genomics, Biomedicine and Translational Medicine, “Iuliu Hațieganu” University of Medicine and Pharmacy, 400337 Cluj-Napoca, Romania; 2“Leon Daniello” Clinical Hospital of Pulmonology, Bronchology Department, 400371 Cluj-Napoca, Romania; 3Department of Surgery, “Iuliu Hațieganu” University of Medicine and Pharmacy, 400337 Cluj-Napoca, Romania

**Keywords:** NSCLC, microRNA, microarray

## Abstract

**Background:** Non-small cell lung cancer (NSCLC) is still one of the types of cancer with the highest death rates. MicroRNAs (miRNAs) play essential roles in NSCLC development. This study evaluates miRNA expression patterns and specific mechanisms in male patients with NSCLC. **Methods:** We report an integrated microarray analysis of miRNAs for eight matched samples of males with NSCLC compared to the study of public datasets of males with NSCLC from TCGA, followed by qRT-PCR validation. **Results:** For the TCGA dataset, we identified 385 overexpressed and 75 underexpressed miRNAs. Our cohort identified 54 overexpressed and 77 underexpressed miRNAs, considering a fold-change (FC) of ±1.5 and *p* < 0.05 as the cutoff value. The common miRNA signature consisted of eight overexpressed and nine underexpressed miRNAs. Validation was performed using qRT-PCR on the tissue samples for miR-183-3p and miR-34c-5p and on plasma samples for miR-34c-5p. We also created mRNA-miRNA regulatory networks to identify critical molecules, revealing NSCLC signaling pathways related to underexpressed and overexpressed transcripts. The genes targeted by these transcripts were correlated with overall survival. **Conclusions:** miRNAs and some of their target genes could play essential roles in investigating the mechanisms involved in NSCLC evolution and provide opportunities to identify potential therapeutic targets.

## 1. Introduction

Lung cancer is an aggressive malignant tumor and the leading cause of cancer-related deaths in both developing and developed countries, being the second most diagnosed cancer worldwide, according to GLOBOCAN 2020 statistics. The statistics for Europe indicate that about half a million cases of lung cancer are diagnosed yearly. This pathology has the highest mortality rate in men and women, causing 384,176 deaths/year [1]. The incidence of lung cancer in European men in 2020 is 315,054 (13.5%), almost double the number among women, 162,480 (7.9%), making the interest in men’s lung cancer genomic profiling attractive, especially for the identification of a specific signature and the discovery of new potential therapeutic approaches that consider sex-related genomic profiles [2].

Based on the histopathological features, lung cancer is broadly classified into two major classes: non-small-cell lung cancer (NSCLC) and small-cell lung cancer (SCLC) [3]. NSCLC includes three main histological subtypes. Lung squamous cell carcinoma (LUSC) and lung adenocarcinoma (LUAD) account for more than 80% of the total cases of lung cancer, and large cell lung cancer (LCLC) completes the rest of up to 20% [4]. Most of these cases (over 75%) are diagnosed in late stages when the therapeutic response is limited and the overall survival poor [5]. In addition, males have an increased risk of death compared to women; males have a reduced benefit from EGFR inhibitors, and additional anti-PD1 inhibitors significantly improved the survival in male patients compared with chemotherapy [6]. Considering these facts, we decided to include only males in our patient cohort.

MicroRNAs (miRNAs) represent a newly discovered class of non-coding RNA molecules consisting of 19–25 nucleotides in length and are a highly conserved type of endogenous non-coding RNA. These transcripts play essential roles in various biological processes such as cell proliferation, differentiation, and apoptosis by binding to the 3′ untranslated region of target mRNAs, thus modulating their expression and altering various signaling pathways [7,8,9]. They are intensively studied due to their capacity to inhibit coding genes that are targeted by the short sequence of miRNAs. Some new studies show that some potential target RNAs of miRNAs are not simply targets of miRNAs but also control miRNA’s function and stability [10].

Functional miRNAs in NSCLC are commonly dysregulated, mainly due to genomic deletion and changes in the methylation status. Thus, these miRNAs may be able to produce changes in many cancer-related pathways and processes (cell cycle, metabolism, epithelial to mesenchymal transition) and modify sensitivity to current therapies [11,12,13].

Several miRNA profiling platforms, from microarray to next-generation sequencing, are available to identify miRNAs prospectively [14]. Microarray technology is a powerful platform for biological exploration. This technology allows the evaluation of coding and several classes of non-coding gene signatures, revealing their importance in tumorigenesis and cancer progression [15,16]. In recent years, microarray technology has been successfully used for miRNA profiling studies to discover new correlations between coding genes and their involved regulatory pathways. The microarray is one of the most consistent techniques in translational medicine, especially when the samples are adequately selected for homogeneity, and the numbers are representative [16,17].

The expression level of different miRNAs can be correlated with several histopathological biomarkers, give us information about the mutational status of genes, and be used as early-cancer detection diagnostic tools [18]. miRNA signatures can also be related to clinical parameters, including disease stage and the presence of targetable mutations, and can predict response to new therapeutic approaches [19,20,21]. Several miRNAs are under investigation for their role as therapeutic targets or modulators of the currently available therapies in NSCLC [22,23,24].

The literature data revealed that miRNAs might show more significant expression divergence between male and female groups in both physiological and pathological conditions [25,26,27,28]. Differences in the incidence and prognosis according to gender were observed in inflammatory lung disease and lung cancer [28,29]. An analysis of the LUAD TCGA cohort identified 73 sex-biased miRNAs (40 male-biased and 33 female-biased miRNAs) [30]. Therefore, this study aims to explore further the sex-specific miRNA expression in a cohort of male NSCLC. A better understanding of the sex-specific miRNA dysregulation can provide essential insights into cancer pathogenesis and better identify new approaches to stratify patients for currently available therapies or identify new cellular vulnerabilities in specific patient cohorts.

Our study focuses on the global miRNA expression patterns evaluated using microarray in male NSCLC paired samples (eight matched samples of male patients with NSCLC, four LUAD and four LUSC, stages IB to IIIA). Our results were compared with the TCGA miRNA datasets. Differentially expressed miRNAs were validated using qRT-PCR in tissue and plasma samples from NSCLC patients. We integrated our miRNA expression results in the ingenuity pathway analysis (IPA) software that was used to identify the underlying regulatory mechanisms and the involved targeted genes. Among the targets, we selected EGFR, IGF-IR, and TGFβ1 proteins for validation on plasma samples using ELISA. Our validation approach uses qRT-PCR and ELISA, two widely available techniques in pathology laboratories which increases the potential integration of our panels in clinical practice.

## 2. Materials and Methods

The study workflow of the present research is presented as a flow chart in Appendix A.

### 2.1. NSCLC TCGA Data Analysis

We performed an analysis on the third level of miRNAs profiling from 467 lung tumors (NSCLC) and 19 normal tumor-adjacent tissues, all from men (Table 1), obtained from the TCGA data portal (https://tcga-data.nci.nih.gov/tcga/), accessed on 21 July 2021. Differential expression analysis was performed using the GeneSpring GX v.13.0 software from Agilent Technologies (Sant Clara, CA, USA). The volcano plot module was applied using a fold change of 1.5 and a *p*-value of < 0.05.

### 2.2. Patient Cohort

Between December 2016 and February 2022, we collected the matched pairs of tumor tissue alongside the normal and peripheral whole blood samples from NSCLC samples, both LUAD and LUSC. The tissue samples were collected using flexible bronchoscopy with endobronchial biopsy, performed under local anesthesia by an experienced operator. Normal tissue for the matched paired samples was collected from the contralateral, healthy lungs of the same patient. This study included patients older than 18 who presented in the Bronchoscopy Department with suspicion of endobronchial lung cancer on imagistic studies (computer tomography or positron emission computed tomography). Exclusion criteria were diagnosis of synchronous or metachronous tumors of other organs, patients with prior oncologic therapy, and the inability to tolerate the procedure. All patients enrolled in the study were informed about the inclusion and exclusion criteria and signed the informed consent. The present study was approved by the institutional ethics committee of Iuliu Hatieganu University of Medicine and Pharmacy (UMPh), No. 438/24 November 2016 and 264/26 June 2018.

We analyzed the paired tumor and normal tissue samples from 8 male NSCLC patients for the microarray profiling to depict dysregulated miRNAs. Of the 8 male patients, 4 were diagnosed with LUAD and 4 with LUSC (Table 2), referred to as UMPh patient cohort. An independent patient cohort was used for qRT-PCR validation of the selected miRNAs at the tissue level consisting of 62 NSCLC patients (28 with LUAD and 34 with LUSC), with their characteristics described in Table 3.

Additional validation was performed at the plasma level in a partially common patient cohort (*n* = 19) for the selected miRNAs and ELISA validation of the EGFR, IGF-IR, and TGFβ1 expression levels (Table 4). The expression levels of the selected miRNAs in plasma samples were compared with the expression levels encountered in the plasma collected from 34 healthy subjects. All the experiments performed in this study included only healthy male controls and NSCLC patients. The healthy control cohort included 34 plasma specimens, with ages ranging from 40 to 61, with an age average of 49.9 years, all males (*n* = 34). These are indispensable as a reference for validating miR-34c-5p at plasma levels.

### 2.3. Sample Processing and Microarray Evaluation

The total RNA extraction and isolation from 8 paired samples (normal and tumoral LUAD tissue) were performed using the TriReagent (Sigma-Aldrich, St. Louis, MO, USA) protocol, followed by the evaluation of RNA concentration using NanoDrop-2000. The microarray protocol used 100 ng of total RNA for each analyzed sample, as this amount is recommended in the standard protocol for miRNA evaluation using miRNA complete labeling and hybridization kit (cat No. 5190–0456 Agilent Technologies, Santa Clara, CA, USA), followed by a purification step using Micro Bio-Spin P-6 Gel Column (Biorad, Hercules, CA, USA, cat No. 732-6221).

Microarray slides (G4870C, 8 × 60 K, miRbase_21, Agilent Technologies) were scanned using SureScan microarray scanner (Agilent Technologies), and then for the extraction of the data, Feature Extraction version 12.0 was used. For the assessment of the altered miRNAs in tumor tissue vs. adjacent normal tissue GeneSpring GX v.13.0 was used applying a fold change (FC) threshold of 1.5 and FDR (false discovery rate) correction (*p*-value ≤0.05). 

### 2.4. Functional Analysis and Target Gene Identification

DIANA miRPath is a specific tool for identifying miRNA-targeted pathway analysis via a web interface (http://www.microrna.gr/miRPathv2, accessed on 28 July 2021) [31]. To determine the target genes most relevant to our miRNAs, we used IPA software [32,33].

### 2.5. miRNA Evaluation of Expression Levels in Tissue and Plasma Samples

Plasma preparation and RNA isolation from NSCLC patients and healthy controls were performed in standard conditions. Peripheral blood on EDTA (ethylenediaminetetraacetic acid) was obtained from each patient after the initial diagnosis and immediately processed (less than two hours after collection) by centrifugation at 4000 rpm for 10 min at room temperature, then aliquoted in 2 mL tubes and deposited at −80 °C.

The extraction of total RNA from plasma samples was performed using the plasma/serum circulating and exosomal RNA purification kit (slurry format), followed by the quantification of RNA concentration using the NanoDrop-1000 spectrophotometer; meanwhile, the RNA was extracted using the TriReagent-based method for tissue samples.

In the microarray experiment, RT-PCR was performed, starting from 50 ng total RNA for both tissue and plasma samples, to test the candidate miRNAs observed to be differentially expressed in a statistically significant manner. Then, RNA was reverse-transcribed using a TaqMan microRNA reverse transcription kit (Applied Biosystem, Foster City, CA, USA, 4366596). RT-PCR reactions were performed on the ViiA7 instrument (Applied Biosystems, Foster City, CA, USA) in a 10μL reaction volume containing TaqMan™ Fast Advanced Master Mix (catalog number: 4444556, Thermo Scientific, Waltham, MA, USA) and specific primers (miR-34c-5p cat No. 000426 and miR-183-3p cat No. 002270, U6 cat No. 001973 and RNU48 cat No. 001006, Life Technologies, Carlsbad, CA, USA).

The expression level was calculated using the 2−ΔΔCt method, U6, and RNU48 for normalization; *p* < 0.05 was considered statistically significant. Additionally, a ROC (receiver operating characteristic) graphical representation was performed to assess the sensitivity and specificity of each evaluated transcript at plasma and tissue levels using GraphPad Prism (https://www.graphpad.com/, Version 6, accessed on 23 October 2022), and the combined ROC curves were generated using the CombiROC online tool [34].

### 2.6. EGFR, IGF-IR, and TGFβ1 Quantification in Serum Samples

The expression levels of EGFR, IGF-IR, and TGFβ1 were encountered in serum samples from NSCLC male patients. Healthy subjects were detected using ELISA, with the Human EGFR DuoSet ELISA (R&D System, Minneapolis, MN, USA, cat No. DY231), Ancillary Reagent Kit 2 (R&D System, cat No. DY008), IGFIR DuoSet ELISA (R&D System, cat No. DY 391), Ancillary Reagent Kit 3 (R&D System, cat No. DY240-05), TGF beta 1 DuoSet ELISA, and Ancillary Reagent Kit 1 (R&D System, cat No. DY 007).

## 3. Results

### 3.1. Clinical and Pathological Characteristics of the Cohorts

The main features of the patients included in the microarray study are listed in Table 2. Equal proportions of males with LUAD (*n* = 4) and LUSC (*n* = 4) were considered, aged 50–79. This assessment included patients from stage IB to IIIA (Table 2). The second cohort of patients (validation cohort), from which tumor tissue and normal tissue were used to validate the selected miRNAs, is described in Table 4. The patients’ age was between 50 and 89 years, with a majority in the 60–69 years range. The selected patients were diagnosed with stage II–IV NSCLC. Moreover, most of these patients were former or current smokers at diagnosis, with only six patients in the LUAD group being never-smokers. The second cohort, used for additional validation, partially overlaps with the first validation cohort, with 12 patients included in both validation cohorts. This cohort included 19 patients from whom plasma samples were collected (Table 4).

### 3.2. Evaluation of Tissue miRNAs’ Expression Levels in NSCLC Patients in TCGA and UMPh Cohort

The differences in miRNA-expression levels in accordance with fold changes (FC) ± 1.5 and a *p* < 0.05 were considered significant for the male samples: 385 overexpressed and 75 underexpressed miRNAs, and for the female samples, 61 overexpressed and 27 downregulated. The TCGA analysis observed substantial differences between male and female NSCLC, particularly in the case of downregulated miRNA, revealing an important miRNA pattern specific for males (63 downregulated and 331 overexpressed) and females (15 downregulated and 9 overexpressed), and only 12 downregulated and 52 overexpressed miRNAs as common signatures (Appendix A) for male and female NSCLC from the altered gene expression signatures.

Additional profiling investigation in our patient cohort favored the identification of the most relevant altered miRNAs based on an analysis of eight paired samples using microarray analysis: lung cancer tumor tissues (TT) versus normal tissues (TN). The microarray analysis identified 77 underexpressed and 54 overexpressed miRNAs in the UMPh male patient cohort.

To further analyze the involvement of these miRNAs in important metabolic and tumorigenic processes, we realized a heatmap analysis, depicted in Figure 1. The heatmap analysis, presenting the altered miRNAs in tumor tissue for the TGCA male NSCLC patient cohort, is summarized in Figure 1A, whereas for the UMPh patient cohort, it is shown in Figure 1B. Figure 1C,D show commonly dysregulated miRNAs between the TCGA and UMPh cohorts for both the underexpressed and overexpressed miRNAs. The nine common underexpressed miRNAs include miR-30a-3p, miR-30a-5p, miR-139-3p, miR-133b, miR-143-3p, miR-34c-3p, miR-34c-5p, miR-34b-3p, and miR-145-5p. The overexpressed eight common miRNAs include miR-183-3p, miR-7-5p, miR-21-5p, miR-1224-5p, miR-194-3p, miR-934, miR-382-5p, and miR-1250-5p.

Additional analyses were performed separately on LUSC and LUAD in both patient cohorts (Appendix A). In the UMPh patient cohort, for LUAD, the study showed 97 altered miRNAs (70 underexpressed and 27 overexpressed); meanwhile for LUSC, we found 171 altered miRNAs (66 underexpressed and 105 overexpressed).

The analysis of the patient cohort for LUAD showed 350 altered miRNAs (85 underexpressed and 265 overexpressed); meanwhile for LUSC, there was observed 522 altered miRNAs (77 underexpressed and 445 overexpressed). Appendix A presents the common signatures among the two patient cohorts - LUAD and LUSC cohorts. Additional common signatures among LUAD and LUSC are shown in Appendix A, separated by upregulated and underexpressed miRNAs overlapping TCGA and our patient cohort dataset.

### 3.3. Functional Analysis and Target Genes Identification

Analysis, using DIANA-miRpath (v.3.0) presented for the commonly altered miRNA signature in male NSCLC patients, reveals significant pathway alterations (Figure 2). Most of the significantly enriched pathways were cancer-associated. These include TP53 signaling, focal adhesion, PI3K-AKT signaling, the Hippo signaling pathway, the TGF-beta signaling pathway, and the adherence junction which were proved to be altered (corrected *p* < 0.05). Additionally, the specific network related to NSCLC emphasized six clear transcripts. Among them, miR-145-5p, miR-34c-5p, and miR-30a-5p were downregulated, and the three upregulated transcripts were miR-21-5p, miR-7-5p, and miR-183-3p, as presented in Figure 3A.

An important issue was related to the targets of the selected miRNAs of the NSCLC pathway performed using KEGG pathways via DIANA-miRPath v3.0, revealing some common targets for both the downregulated and overexpressed transcripts that can be observed from the Circos graphical representation (Figure 3B). This is the case in the *NRAS* gene that is targeted by all three overexpressed transcripts and one downregulated miRNA (miR-30a-5p). *CCND1* is targeted by two overexpressed miRNAs (miR-21-5 and miR-7-5p) and two downregulated ones (miR-145-5p and miR-30a-5p).

### 3.4. IPA miRNA–Gene Regulatory Network on Male NSCLC

The miRNA–gene regulatory network in Figure 4 and Appendix A illustrates the interconnection between the miRNAs and the pathways these transcripts are involved in. This network offers a comprehensive image of altered miRNA signatures and the signaling pathways disturbed in male patients with an NSCLC diagnosis. The main challenge is identifying the most relevant altered miRNAs and target genes, considering the common and specific target genes. This approach can have an essential role in developing novel targeted therapies. miR-21, miR-34c-5p, miR-30a-3p, miR-143-3p, and miR-145-5p can be considered candidates considering the higher number of genes targeted. These transcripts are directly and indirectly connected with the essential genes correlated with overall survival, such as TGFB1, p38 MAPK, THEMIS, and SMAD2, as observed in the IPA network analysis (Figure 4A); the core element of the network is miR-34c-5p. Figure 4B presents the miRNA network cellular movement, development, growth, and proliferation as interconnected with the essential genes and able to predict overall survival in male lung cancer (*ILF3*, *IGF1R*, and *PAX3*); in this network can be observed a direct interconnection between miR-34c-5p and miR-34c-3p with *PAX3-FOXO1* or IGF1R.

### 3.5. RT-PCR Tissue Validation

Given the context of the information displayed in Figure 4, two miRNAs were chosen for further validation based on the statistically significant FC obtained from our microarray data and the common signature with the TCGA data. Therefore, we set one representative from the miR-34 family members, miR-34c-3p, and miR-35c-5p; these two miRNAs are in the top ten downregulated transcripts in NSCLC. In addition, in the case of the upregulated miRNAs, from the common signature, we selected miR-183-3p (one of the top five upregulated transcripts in the UMPh patient cohort and in the top 25 upregulated miRNAs in the TCGA data set in NSCLC). The validation at tissue level was performed in an independent tissue patient cohort comprising 28 matched paired LUAD samples and 34 cases matched paired LUSC samples (Figure 5). These transcripts are not affected by different cancer stages, as observed from the TCGA dataset (Appendix A) and our patient cohort (Appendix A).

### 3.6. RT-PCR Plasma Validation

The expression of miR-34c-5p was upregulated in the plasma samples of cancer patients compared with the healthy controls (*p* = 0.004 and AUC = 0,8467) and are concordant with those observed in the microarray experiment in both the tumor tissue and plasma samples (Figure 6), in 19 NSCLC plasma samples and 34 healthy males used as controls.

### 3.7. EGFR, IGF-IR, and TGFβ1 Quantification in Serum Male NSCLC Patients

As an additional validation step, we evaluated circulating EGFR, IGF-IR, and TGFβ1 in male NSCLC patients (*n* = 19) and healthy subjects (*n* = 34). The results are shown in Figure 7. The results indicate decreased concentration levels in the serum samples for EGFR in male NSCLC patients versus healthy subjects. No statistically significant alteration was observed for IGF-IR and TGFβ1.

## 4. Discussion

To identify the miRNA candidates as potential therapeutic targets, we overlapped our microarray data with a male NSCLC TCGA dataset containing the miRNA expression profiles from the tumor tissue, followed by validation at the plasma level of one miRNA candidate (miR-34c-5p) and the EGFR level of expression in the serum samples.

However, given that a large number of miRNAs can target a single mRNA and a single miRNA can target several mRNAs, further studies are needed to better understand the gender-associated differences in male NSCLC pathways and emphasize the complex regulatory networks, as revealed in Figure 3. Our bioinformatics network analysis supports the previously presented idea that mRNAs are targeted by multiple miRNAs [35]. This is the case for NRAS, EGFR, CDK6, and CCND1, as shown in the regulatory pathways in Figure 3A.

The expression of miR-34 in lung cancer has also been evaluated in several studies indicating that the world of three miR-34 family members is decreased in lung tumor tissue compared to normal tissues [36,37,38]. This family also includes miR-34c-3p and miR-34c-5p, and these transcripts have a reduced expression according to the analysis performed in our study and the same seed sequence of miRNA as miR-449 [39,40]. Another study revealed that the low expression of circulating miR-34 family members correlates with poor survival in NSCLC patients. The MiR-34 family members act as tumor suppressors in NSCLC [41,42] and are considered critical regulators of the TP53 signaling [43] and EMT (epithelial to mesenchymal transition) [44]. A recent study revealed that miR-34b and miR-34c are more effective tumor suppressors than miR-34a [40], which is directly related to the TP53 signaling [45].

Moreover, miR-34a/c appears to be involved in TRAIL-induced apoptosis in lung cancer [36]. The limitations in our study are related to the reduced number of samples, particularly in the plasma quantification of miR-34c-5p. As can be observed in Figure 8, this transcript is involved in all steps of tumor progression directly or indirectly, together with other multiple biological processes.

MiR-183 is a transcript significantly upregulated in NSCLC, suggesting that it has an oncogenic function in the lung cancer pathogenesis [46]. Our study confirmed the overexpression in tumor tissue versus normal tissue from NSCLC patients versus healthy controls. MiR-183 is related to cancer progression by repressing the expression of PTEN [46] and PIK3CA [47]. The PIK3CA regulatory axis is identified as a potentially effective therapeutic strategy for lung cancer [47], considering that this signaling pathway is also targeted by other significant miRNAs, such as miR-21-5p and miR-34c (Figure 2). It is essential to mention that these altered transcripts are interconnected with genes correlated with overall survival in lung cancer, as shown, highlighted in red, in Figure 3A. The MiR-34 family members and miR-21 demonstrated the highest degree of connectivity, illustrated in the complex miRNA regulatory network (Figure 4A).

EGFR represents an essential element of this NSCLC network. EGFR is a protein involved in various pathways that control cell proliferation, differentiation, apoptosis, angiogenesis, and metastasis, including Ras/Raf/MAPK, JAK/STAT, and PI3K-Akt pathways [48]. Moreover, tyrosine kinase inhibitors (TKIs) targeting EGFR have been developed with significant improvements in response rates and survival [49,50]. IGF-1R was reported as regulating apoptosis and its overexpression in tumor tissues contributing to an antiapoptotic effect by improving cell survival, with its activation resulting in transmitting signals downstream through the PI3K-AKT1-MTOR and MAPK pathways [51]. Cross-talk between EGFR and IGF-1R pathways promotes resistance to EGFR TKIs and monoclonal antibodies for EGFR and the upregulation of IGF-1R in NSCLC patients, inducing EGFR TKI resistance [52,53,54]. Regarding TGFβ1, this transcript was appraised given its well-known involvement in EMT, including NSCLC [55]. The PI3K-Akt and RhoA pathways are activated following TGFβ1 signaling [56].

As previously reported, miR-21 is upregulated under conditions in which EGFR signaling is activated; a fact proved in a cohort of non-smoking patients [57]. Based on our data, it was observed that EGFR is targeted by overexpressed miR-7-5p and downregulated miR-145-5p and miR-30a-5p. Reports have proven that the serum EGFR levels are connected with aggressive cancer development. Quantifying serum/plasma EGFR levels in NSCLC patients and the correlation with clinical parameters remains controversial. In our patient cohort, the expression levels were downregulated in NSCLC patients versus the healthy control group. Increased expression levels of EGFR were observed in advanced stages, and decreased expression levels were observed post-surgery [58]. EGFR proved to be dysregulated in male NSCLC; the importance of this potential sex-related tumor particularity remains to be determined.

## 5. Conclusions

This study demonstrates the central role of miRNAs in male NSCLC cancer pathogenesis. Our study identified nine underexpressed and eight overexpressed miRNAs that can be further investigated to better understand the particularities of male NSCLC. miRNA panels can complement the currently available diagnostic approaches represented by pathology and the associated molecular pathology methods by allowing a better stratification of patients in terms of disease management, treatment selection, and prognosis. Based on the miRNA panel expression profiles, we can get a dynamic image of the associated tumor biological processes that will identify patients that need additional adjuvant therapies depending on their stage at diagnosis. Our data reveal that the regulatory networks in male NSCLC can be relied on, as this will identify the most relevant connections between miRNA and associated targets.

In conclusion, in the present study, we showed the involvement of specific miRNAs in male NSCLC development. The miRNA–mRNA regulatory networks identified in male NSCLC can be used in additional studies that can further decode the sex-specific particularities of this cancer. Moreover, the plasma validation of miR-34c-5p expression could add to the value of this circulating biomarker in NSCLC. The small sample size is an explicit limitation of this study, with inquiries remaining to be answered before moving further with these data into clinical practice.

## Figures and Tables

**Figure 1 jpm-12-02056-f001:**
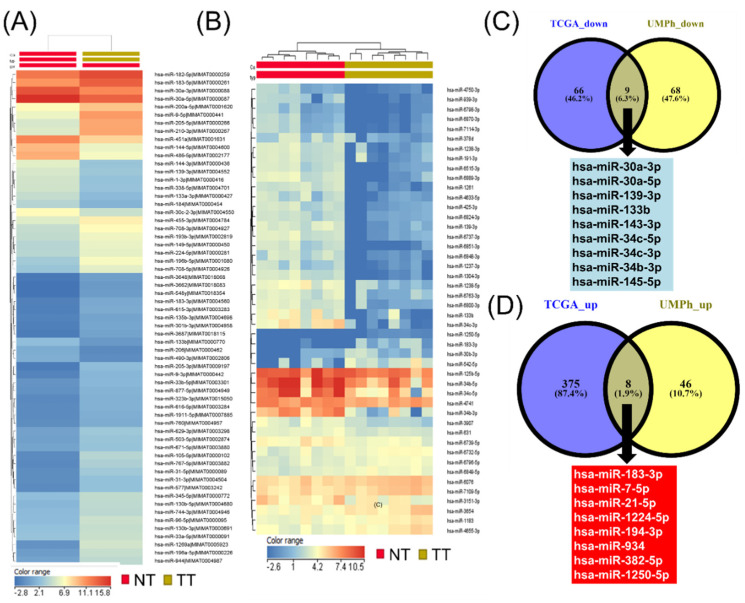
miRNA profiling in NSCLC male patients. (**A**) Heatmap for the TGCA dataset. (**B**) Heatmap for UMPh patient cohort (the blue color denotes underexpression, whereas the red color suggests overexpression). (**C**) Venn diagram for underexpressed miRNAs between TCGA (blue) and UMPh (yellow) cohorts. (**D**) Venn diagram for overexpressed miRNAs between TCGA (blue) and UMPh (yellow) cohorts.

**Figure 2 jpm-12-02056-f002:**
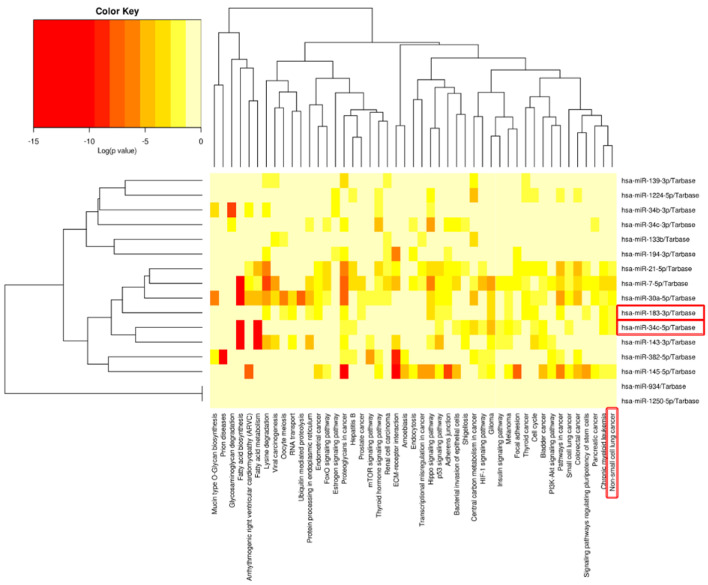
Heatmaps of significant pathways predicted by DIANA-miRPath (v.3.0) for specific common miRNAs signature in male NSCLC patients, with clustering based on significance levels. Paths are represented on the x-axis and miRNAs on the y-axis. The color code expresses the log (*p*-value), with the most relevant predicted miRNA–pathway interactions shown in red.

**Figure 3 jpm-12-02056-f003:**
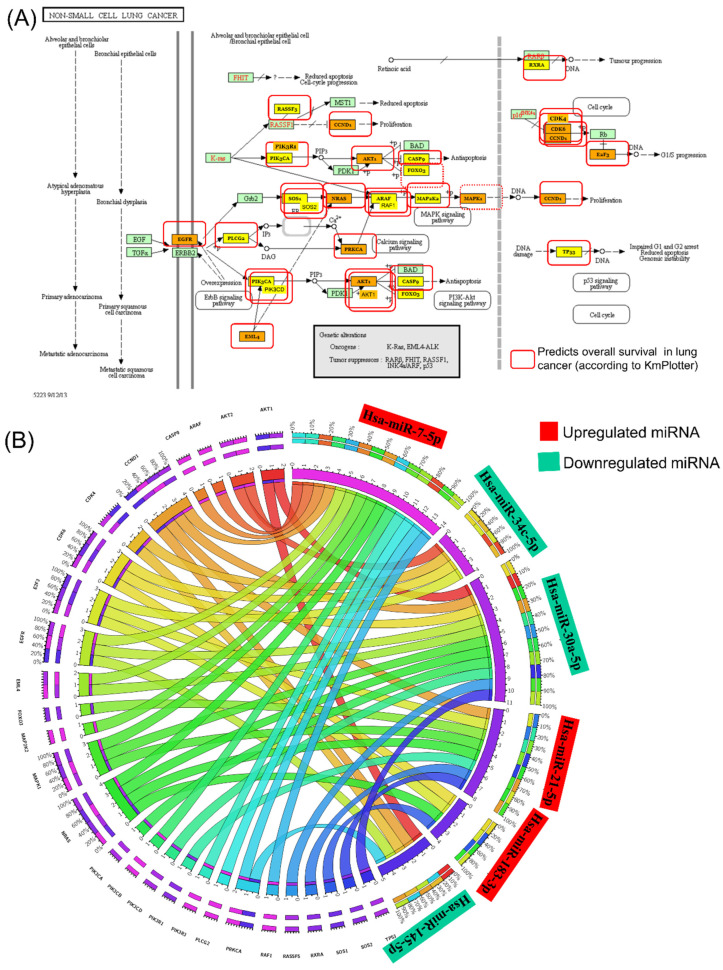
NSCLC signaling. (**A**) NSCLC network performed using KEGG pathways via DIANA-miRPath v3.0. Yellow are genes targeting one miRNA, and orange are genes targeting at least two miRNAs. Red squares mark the genes correlated with the overall survival of NSCLC patients using the KM plotter online tool. (**B**) Circos plot emphasizes the interconnection with the downregulated transcripts (miR-145-5p, miR-34c-5p, and miR-30a-5p) and three upregulated transcripts (miR-21-5p, miR-7-5p, and miR-183-3p) and critical genes related to this signaling.

**Figure 4 jpm-12-02056-f004:**
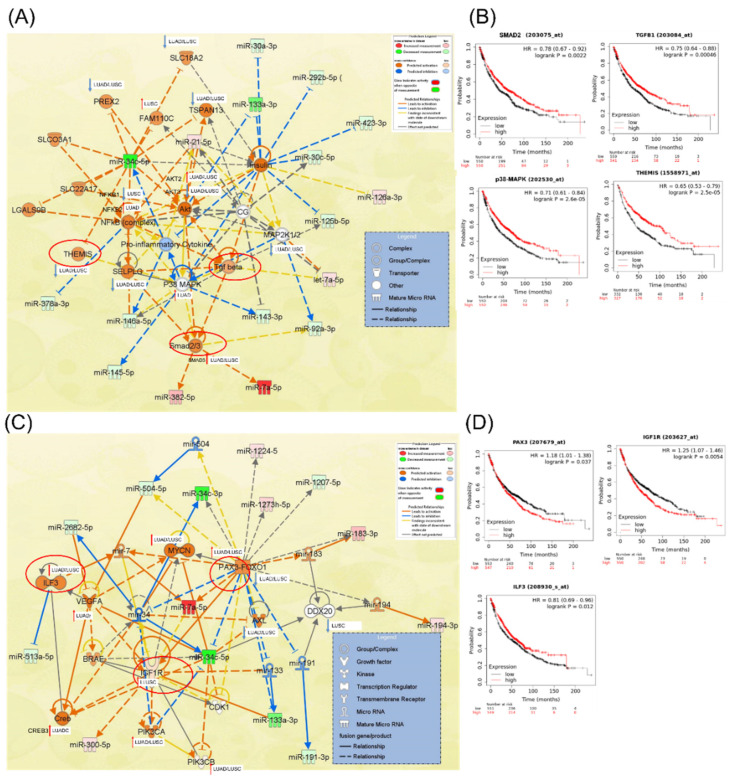
IPA network for altered miRNAs in UMPh patient cohort. (**A**) Network related to cancer, organismal injury, abnormalities, and reproductive system disease; ↑upregulated genes, ↓downregulated genes in LUAD or LUSC according to UALCAN (http://ualcan.path.uab.edu). (**B**) Genes from the network correlated with overall survival in NSCLC, generated using Kaplan–Meier plotter online platform (https://kmplot.com/analysis/ accessed on 23 October 2022). (**C**) Network related to cellular movement, development, growth, and proliferation. (**D**) Genes from the network correlated with overall survival in NSCLC, generated using Kaplan–Meier plotter online platform (https://kmplot.com/analysis/ accessed on 23 October 2022).

**Figure 5 jpm-12-02056-f005:**
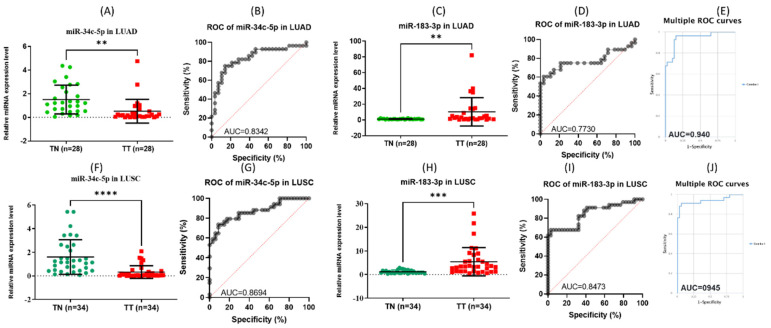
Tissue qRT-PCR data validation for miR-34c-5p and miR-183-3p in NSCLC. (**A**) miR-34c-5p expression levels in LUAD. (**B**) ROC curve for miR-34c-5p in LUAD. (**C**) miR-183-3p expression levels in LUAD. (**D**) ROC curve for miR-183-3p in LUAD. (**E**) Combined ROC curves for miR-34c-5p and miR-183-3p in LUAD, generated using CombiROC online tool. (**F**) miR-34c-5p expression levels in LUSC. (**G**) ROC curve for miR-34c-5p in LUSC. (**H**) miR-183-3p expression levels in LUSC. (**I**) ROC curve for miR-183-3p in LUSC. (**J**) Combined ROC curves for miR-34c-5p and miR-183-3p in LUSC, generated using CombiROC online tool (** *p* ≤ 0.01, *** *p* ≤ 0.001, **** *p* ≤ 0.0001).

**Figure 6 jpm-12-02056-f006:**
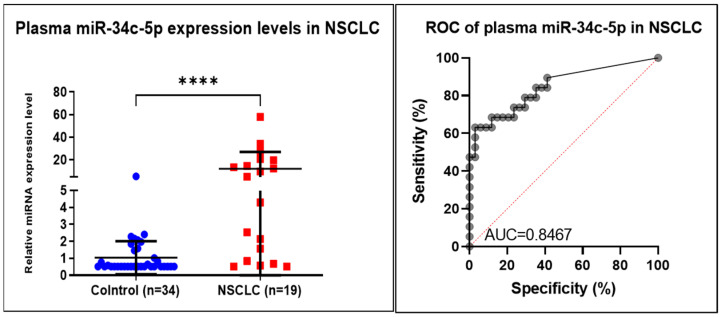
Plasma qRT-PCR data validation of the miR-34c-5p in NSCLC (*n* = 19) and healthy controls (*n* = 34), and receiver operating characteristic (ROC) curve AUC = 0.8467 (**** *p* ≤ 0.0001).

**Figure 7 jpm-12-02056-f007:**
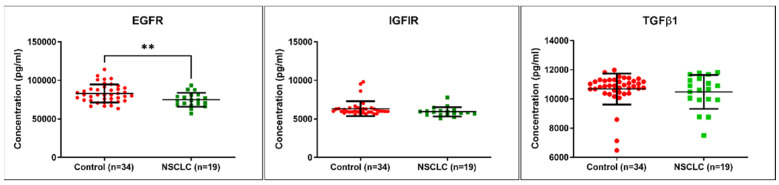
EGFR, IGF-IR, and TGFβ1 quantification in serum of male NSCLC patients (*n* = 19) and healthy subjects (*n* = 34) using ELISA (** *p* ≤ 0.01).

**Figure 8 jpm-12-02056-f008:**
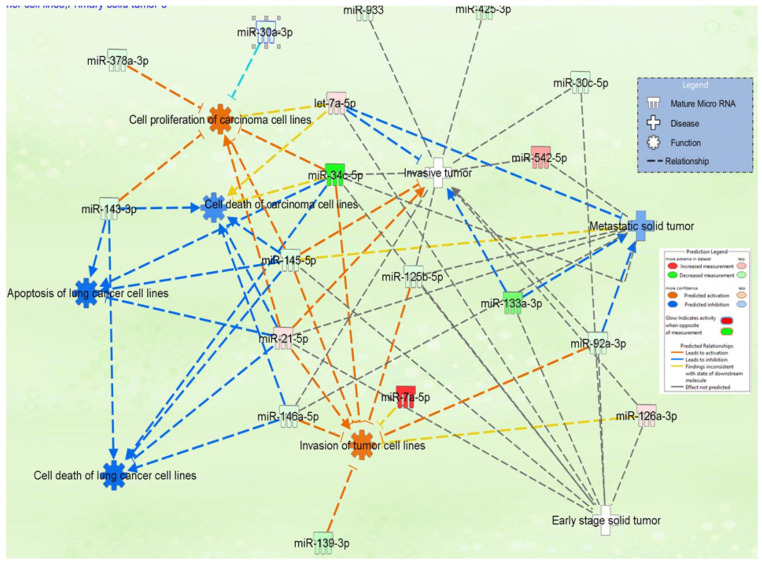
IPA network emphasizes the interconnection of miR-34c-5p with key cancer hallmarks, including invasion, proliferation, and metastatic potential.

**Table 1 jpm-12-02056-t001:** Male NSCLC TCGA samples’ characteristics.

Samples Parameters		LUAD (*n* = 210)	LUSC (*n* = 257)
Age	Median, Range ♂	67, 41–88	68, 41–90
	Unknown	9	4
T stage	T1	56	52
	T2	121	153
	T3	23	43
	T4	9	9
	Tx	1	-
N stage	N0	131	168
	N1	48	63
	N2	28	21
	N3	-	-
	Nx	2	5
	N unknown	1	-
M stage	M0	142	199
	M1	12	3
	Mx	55	55
	M unknown	1	-
Tumor stage	I	99	117
	II	63	95
	III	32	39
	IV	12	3
	Unknown	4	3
Smoking status	Never smoker	19	6
	Current smoker	61	86
	Quit > 15 years	63	39
	Quit ≤ 15 years	57	113
	Quit (unknown)	3	4
	Unknown	7	9

**Table 2 jpm-12-02056-t002:** Sample characteristics of the 8 male NSCLC patients used for miRNA microarray evaluation.

Demographics		LUAD *n* = 4	LUSC *n* = 4
		No. of Patients (%)	No. of Patients (%)
Age	50–59	1 (25)	0 (0)
	60–69	2 (50)	3 (75)
	70–79	1 (25)	1 (25)
Sex	M	4 (100)	4 (100)
Stage	IB	0 (0)	1 (25)
	IIA	2 (50)	1 (25)
	IIB	1 (25)	2 (50)
	IIIA	1 (25)	0 (0)
Smoking status	Never smoker	2 (50)	0 (0)
	Former smoker	0 (0)	2 (50)
	Current smoker	2 (50)	2 (50)

**Table 3 jpm-12-02056-t003:** Clinicopathological characteristics of the 62 NSCLC samples used for the miRNA expression validation with qRT-PCR validation.

Characteristics		LUAD *n* = 28	LUSC *n* = 34
		No. of Patients (%)	No. of Patients (%)
Age	50–59	8 (28.6)	9 (26.5)
	60–69	13 (46.4)	14 (41.1)
	70–79	6 (21.4)	9 (26.5)
	80–89	1 (3.6)	2 (5.9)
Sex	M	28 (100)	34 (100)
T	T2	3 (10.7)	3 (8.8)
	T3	10 (35.7)	9 (26.5)
	T4	15 (53.6)	22 (64.7)
N	N0	4 (14.3)	3 (8.8)
	N1	7 (25)	6 (17.6)
	N2	14 (50)	21 (61.7)
	N3	3 (10.7)	4 (11.8)
M	M0	12 (42.8)	26 (76.5)
	M1	16 (57.2)	8 (23.5)
Stage	II	2 (7.2)	2 (5.9)
	III	10 (35.7)	25 (73.5)
	IV	16 (57.1)	7 (20.6)
Smoking status	Never smoker	6 (21.4)	0 (0)
	Current smoker	13 (46.4)	17 (50)
	Former smoker	9 (32.2)	17 (50)

**Table 4 jpm-12-02056-t004:** Demographic and histopathological diagnosis of the 19 NSCLC patients used for plasma qRT-PCR validation of miR-34c-5p and miR-183-3p, and ELISA determination for EGFR, IGF-IR, and TGFβ1 from serum.

Characteristics		LUAD *n* = 4	LUSC *n* = 15
		No. of Patients (%)	No. of Patients (%)
Age	50–59	2 (50)	5 (33.3)
	60–69	1 (25)	5 (33.3)
	70–79	1 (25)	4 (26.6)
	80–89	0 (0)	1 (6.7)
Sex	M	4 (100)	15 (100)
Stage	II	1 (25)	0 (0)
	III	2 (50)	12 (80)
	IV	1 (25)	3 (20)
T	T2	3 (75)	2 (13.3)
	T3	1 (25)	3 (20)
	T4	0 (0)	10 (66.7)
N	N0	2 (50)	0 (0)
	N1	0 (0)	2 (13.3)
	N2	1 (25)	11 (73.4)
	N3	1 (25)	2 (13.3)
M	M0	3 (75)	11 (73.4)
	M1	1 (25)	4 (26.7)
Smoking status	Never smoker	1 (25)	0 (0)
	Current smoker	1 (25)	7 (46.7)
	Former smoker	2 (50)	8 (53.3)

## Data Availability

Please revise according to comment.

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
