# Peer review of "Identification of Potential microRNA Panels for Male Non-Small Cell Lung Cancer Identification Using Microarray Datasets and Bioinformatics Methods"

_jpm, 2022, doi:10.3390/jpm12122056_

Round 1

Reviewer 1 Report (Previous Reviewer 2)

Revisions of the grammar and sentence structure are still needed.

Author Response

The authors of the present manuscript want to express their gratitude toward the Reviewers who contributed significantly to the article's improvement through their expertise and suggested modifications. Therefore, every comment in the revision was addressed (and marked in red in the revised version), and the article was modified according to the Reviewer’s instructions. The modifications are highlighted in red.

Corrections are done, highlighted in red in the text ( using track changes)

Reviewer 2 Report (New Reviewer)

       This manuscript examined the roles of miRNAs in male NSCLC cancer pathogenesis and revealed the miRNA-mRNA regulatory networks for the same. In addition, plasma validation of miR-34c-5p expression could add to the value of this circulating biomarker in NSCLC. However, a few issues must be substantially addressed in the revised manuscript.

1.      The quality of all figures should be improved, especially Figure 1. The font size is too small for readers.

2.      It is written in the Sample processing and microarray evaluation that applying a fold change (FC) threshold of > 1.5 and applied FDR (False Discovery Rate) correlation (p-value ≤0.05) but in the results part, the p-value < 0.05 were considered significant. Which one is correct? Please check throughout the manuscript.

3.      Are the identified miRNA expression patterns of NSCLC patients associated with any particular characteristics?

4.      From the miRNA regulatory mechanisms and the involved targeted genes, the author selected EGFR, IGF-IR, and TGFβ1 proteins for validating plasma samples using ELISA. Please state the reason and criteria for selection.

5.      Why does this study evaluate miRNA expression patterns and their mechanisms specific to male NSCLC patients? If the reason was to identify the sex-related genomic profile, then you should have focused on the male and female NSCLC patients.

Author Response

Response to reviewers

The authors of the present manuscript want to express their gratitude toward the Reviewers who contributed significantly to the article's improvement through their expertise and suggested modifications. Therefore, every comment in the revision was addressed (and marked in red in the revised version), and the article was modified according to the Reviewer’s instructions. The modifications are highlighted in red.

Reviewer 2

Comments and Suggestions for Authors

     This manuscript examined the roles of miRNAs in male NSCLC cancer pathogenesis and revealed the miRNA-mRNA regulatory networks for the same. In addition, plasma validation of miR-34c-5p expression could add to the value of this circulating biomarker in NSCLC. However, a few issues must be substantially addressed in the revised manuscript.

  1. The quality of all figures should be improved, especially Figure 1. The font size is too small for readers.

All the figures will be saved as high resolution and will be sent it as separate files.

  1. It is written in the Sample processing and microarray evaluation that applying a fold change (FC) threshold of > 1.5 and applied FDR (False Discovery Rate) correlation (p-value ≤0.05) but in the results part, the p-value < 0.05 were considered significant. Which one is correct? Please check throughout the manuscript.

Correction for FC threshold as ±1.5.

  1. Are the identified miRNA expression patterns of NSCLC patients associated with any particular characteristics?

We emphasized miRNA expression patterns of male NSCLC patients, we didn’t correlate with any other characteristics, as can be observed from Figures S5 and S6.

Paragraph extracted from the text: “These transcripts are not affected by different cancer stages, as observed from the TCGA dataset (Figure S5) or our patient cohort (Figure S6).”

  1. From the miRNA regulatory mechanisms and the involved targeted genes, the author selected EGFR, IGF-IR, and TGFβ1 proteins for validating plasma samples using ELISA. Please state the reason and criteria for selection.

We added the following paragraph underlining the processes and signalling pathways in which EGFR, IGF-IR, and TGFβ1 are involved, highlighting the involvement of EGFR and IGF-1R in resistance to EGFR TKI inhibitors: ” EGFR is a protein involved in various pathways that control cell proliferation, differentiation, apoptosis, angiogenesis and metastasis, including  Ras/Raf/MAPK, JAK/STAT and PI3K-Akt pathways (48). Moreover, tyrosine kinase inhibitors (TKIs) targeting EGFR have been developed with significant improvements in response rates and survival (49, 50). IGF-1R was reported as regulating apoptosis, its overexpression in tumor tissues contributing to an antiapoptotic effect by improving cell survival, and its activation resulting in transmiting signals downstream through the PI3K-AKT1-MTOR pathway and through the MAPK pathway (51). Cross-talk between EGFR and IGF-1R pathways promotes resistance to EGFR TKIs and monoclonal antibodies for EGFR, upregulation of IGF-1R in NSCLC patients inducing EGFR TKI resistance (52-54). Regarding TGFβ1, this transcript was appraised given its well known involvement in EMT, including NSCLC (55). Among others, PI3K-Akt and RhoA pathways are activated following TGFβ1 signaling (56).”

Additional network representation was added as a supplementary file to emphasize the interconnection of the key altered miRNA with these genes.

Figure S4. IPA network emphasis the interconnection of miR-34 family members and miR-185-5p, with emphasis on key elements EGFR, IGF1R and TGFB1 (marked with the red circle), transcripts selected for additional validation at the protein level by ELISA

  1. Why does this study evaluate miRNA expression patterns and their mechanisms specific to male NSCLC patients? If the reason was to identify the sex-related genomic profile, then you should have focused on the male and female NSCLC patients.

The study focused on male NSCLC patients considering the high incidence versus women and high mortality rate in males versus women. Indeed, a study comparing the genomic profiling of male vs female patients would be interesting and needed in NSCLC. Therefore, we intend to realize a new study by evaluating the genomic profile of females vs. males in age/stage-matched samples.

As it was previously suggested, we added, as supplementary data, the TGCA analysis separately on male and females to support our hypothesis.

Paragraph added in the text:

Based on the histopathological features, lung cancer is broadly classified into two major classes: non-small-cell lung cancer (NSCLC) and small-cell lung cancer (SCLC) (3). NSCLC includes three main histological subtypes: lung squamous cell carcinoma (LUSC), and lung adenocarcinoma (LUAD); these account for more than 80% of the total cases of lung cancer and large cell lung cancer (LCLC), completing the rest of up to 20% (4). Most of these cases (over 75%) are diagnosed in late stages when the therapeutic response is limited and the overall survival poor (5). Also, males have an increased risk of death compared to women; males have a reduced benefit from EGFR inhibitors, and additional anti-PD1 inhibitors significantly improved the survival in male patients compared with chemotherapy(6). Considering these facts, we decide to include only males in our patient cohort.

This manuscript is a resubmission of an earlier submission. The following is a list of the peer review reports and author responses from that submission.

Round 1

Reviewer 1 Report

In this study, the authors perform an integrated analysis of miRNAs for 8 matched samples of male patients with NSCLC (4 lung adenocarcinoma, 4 lung squamous cell carcinoma) and of matched samples from the TCGA database. Comparison of miRNA profiling of these cohorts revealed 9 miRNAs that were underexpressed in both cohorts and 8 miRNAs that were overexpressed in both cohorts. Functional analysis with DIANA-miR-path and KEGG pathways revealed common targets for downregulated and overexpressed transcripts. Expression levels of target genes of commonly upregulated and downregulated miRNAs correlated with overall survival. To validate the microarray data, expression levels of two miRNAs were measured using qRT-PCR in an independent patient tissue cohort and plasma cohort.

Overall, the manuscript is well written. However, there are multiple concerns regarding the manuscript, which are detailed below:

Major concerns:

  1. Why were only males with NSCLC included in this study? The authors do not provide explanation or rationale for only studying males.
  2. The miRNA analysis was performed on matched tissue samples from patients with NSCLC. Although alterations in expression of miRNAs in tumor tissues is intriguing, the authors provide no evidence that miRNA profiling will add to information that is obtained from histopathological analysis, clinicopathological staging, and next-generation sequencing from tumor tissue samples.
  3. The authors fail to provide a detailed description regarding how matched tumor and normal lung tissues from NSCLC patients in the UMPh cohort were obtained. Were these NSCLC patients who underwent surgical lung resections, similar to the TCGA cohort?
  4. Fig S1 shows expression levels of target genes of commonly downregulated and upregulated miRNAs are correlated with overall survival in lung cancer patients. However, other miRNAs also regulate expression of these target genes. The authors fail to show that expression of these genes are affected by downregulation or upregulation of the miRNAs they identify in tumor samples.
  5. Fig. 5: the validation of miR-34c-5p expression in plasma samples compared 19 NSCLC patients and 37 healthy males as controls. The authors fail to provide any detail regarding the subjects in the control cohort. Are there differences in age and smoking status between the NSCLC and control cohorts, which may contribute to differences in miR-34c-5p expression?
  6. Authors suggest that plasma miR-34c-5p expression levels could be used as a biomarker for NSCLC diagnosis. However, in the plasma validation cohort, 18 out of 19 patients had advanced stage NSCLC (stage III-IV), and none of the patients had stage I disease. Thus, whether miR-34c-5p expression levels in the plasma can be used to diagnose early-stage NSCLC is unknown.

Minor concerns:

  1. There are several typos. For example, on Page 7, line 188: Figures 1C and 1D show common miRNAs between TCGA and UMPh, not common genes as indicated in the text. Figure 1 legend: (D) is a Venn diagram, not a heatmap for overexpressed miRNAs. I recommend careful proofreading of the text and figure legends by a native English speaker/writer.

Reviewer 2 Report

The purpose of Dr. Ioana Berindan-Neagoe’s work was to evaluates miRNA patterns and specific mechanisms in male patients with NSCLC. They overlapped their microarray data with a male NSCLC TCGA dataset containing the miRNA expression profiles from the tissue, followed by validation at the plasma level of one miRNA candidate (miR-34c-5p). They further evaluated circulating EGFR, IGF-IR, and TGFβ1 in NSCLC male patients and healthy subjects as an additional validation step. The Authors should have used a more up to date technology to evaluate human miRNAs.

Moreover the following major issues are present: 

1. Why the patient cohorts only include male NSCLCs, female NSCLC patients are needed to better understand gender-associated differences.

2. Please explain in detail the inclusion and exclusion criteria.

3. Please explain why choose miR-183-5p, miR-34c-5p, and miR-30a-5p.

4. Is there a mistake with“8 downregulated and 9 overexpressed miRNAs”in Abstract Section, but “The underexpressed 9 common miRNAs… The overexpressed 8 common miRNAs…”in Results Section.

5. This manuscript is replete with verbose language, revisions of the grammar and sentence structure are needed.
